# Preservation versus resection of Denonvilliers' fascia in total mesorectal excision for male rectal cancer: follow-up analysis of the randomized PUF-01 trial

Jiafeng Fang [1,14], Bo Wei[1,14], Zongheng Zheng[1,14], Jian'an Xiao[2], Fanghai Han[3], Meijin Huang[4], Qingwen Xu[5], Xiaozhong Wang[6], Chuyuan Hong[7], Gongping Wang[8], Yongle Ju[9], Guoqiang Su[10], Haijun Deng [11], Jinxin Zhang[12], Jun Li[1], Xiaofeng Yang[1], Tufeng Chen[1], Yong Huang[1], Jianglong Huang[1], Jianpei Liu[1], Hongbo Wei [1] ✉ & Chinese Postoperative Urogenital Function (PUF) Research Collaboration Group[13]*

Traditional total mesorectal excision (TME) for rectal cancer requires partial resection of Denonvilliers' fascia (DVF), which leads to injury of pelvic autonomic nerve and postoperative urogenital dysfunction. It is still unclear whether entire preservation of DVF has better urogenital function and comparable oncological outcomes. We conducted a randomized clinical trial to investigate the superiority of DVF preservation over resection (NCT02435758). A total of 262 eligible male patients were randomized to Laparoscopic TME with DVF preservation (L-DVF-P group) or resection procedures (L-DVF-R group), 242 of which completed the study, including 122 cases of L-DVF-P and 120 cases of L-DVF-R. The initial analysis of the primary outcomes of urogenital function has previously been reported. Here, the updated analysis and secondary outcomes including 3-year survival (OS), 3-year disease-free survival (DFS), and recurrence rate between the two groups are reported for the modified intention-to-treat analysis, revealing no significant difference. In conclusion, L-DVF-P reveals better postoperative urogenital function and comparable oncological outcomes for male rectal cancer patients.

Rectal cancer (RC) is one of the most common malignant tumors in the world[1]. Surgical resection is the chief therapeutic method for RC, and since first proposed by R.J. Heald in 1982, total mesorectal excision (TME) has been widely applied for mid-low rectal cancer (MLRC)[2]. TME greatly reduces local recurrence and improves long-term survival, thus has been generally considered as a standard surgical principle for MLRC[3]. However, due to intraoperative injury of the pelvic autonomic nerve (PAN)[4,5], the incidence of urogenital dysfunction after TME

surgery stays high and largely affects quality of life[6,7]. Thus, preservation of PAN during TME surgery has gained great attention from surgeons and patients.

Traditional TME surgery required dissection anterior to Denonvilliers' fascia (DVF) and thus DVF should be partly resected[8]. However, subsequent studies revealed that resection of DVF may probably lead to injury of PAN, thus DVF should be well preserved if possible[9,10]. Nevertheless, TME with DVF preservation was doubted and

A full list of affiliations appears at the end of the paper. *A list of authors and their affiliations appears at the end of the paper.
✉e-mail: weihb@mail.sysu.edu.cn

restrictedly applicated, because it used to be considered no surgical plane posterior to DVF, as well as uncertain oncological safety of DVF preservation[8,11]. Thus, traditional TME with partly DVF resection was still routinely performed in many large-scale medical centers regardless of tumor location and extent.

With studies on both cadavers and surgical videos, we demonstrated a surgical landmark line for intraoperative identification of DVF[12,13]. Regardless of mobilization of the peritoneal reflection, dissection below this marker line resulted in easy entry posterior to DVF. Thus, dissection posterior to DVF and the entire preservation of DVF became practicable[14,15]. However, although some surgeons also agreed that DVF preservation could be performed for tumors located on the dorsal or dorsolateral side, whether preservation of DVF has better postoperative urogenital function and similar oncological outcomes compared with partly DVF resection is still unclear, indicating that conducting clinical trials to prove the superiority of DVF preservation is in urgent need.

Thus, we, together with the Chinese Postoperative Urogenital Function (PUF) Research Collaboration Group, conducted a prospective, multicenter, randomized clinical trial (PUF-01) to evaluate the safety and effect of DVF preservation during laparoscopic TME on postoperative urogenital function protection and oncological safety in male patients with MLRC (www.ClinicalTrial.gov, registration: NCT02435758). The initial short-term results revealed that compared with DVF resection (L-DVF-R), DVF preservation (L-DVF-P) presented great advantages in lower incidences of urinary, erectile, and ejaculation dysfunctions, while similar surgical outcomes[16]. Nevertheless, whether preservation of DVF will impact the oncological outcome is still largely elusive and thus limits its application.

In this work, with the updated analysis of urogenital function and follow-up oncological outcomes of the PUF-01 trial, we investigate both function protection and oncological safety of DVF preservation during laparoscopic TME for male rectal cancer patients. The per-protocol analysis reveals that the postoperative urogenital function was better in the L-DVF-P group. The modified intention-to-treat analysis for oncological data reveals no significant differences in 3-year overall survival (OS), 3-year disease-free survival (DFS), and recurrence rate between the two groups. Taking together, L-DVF-P reveals better postoperative urogenital function and comparable oncological outcomes for male rectal cancer patients.

## Results

### Study population
From August 26, 2015, through May 6, 2020, a total of 262 patients were enrolled and randomly assigned to the Exp-group or Con-group (n = 131, respectively). As shown in Fig. 1, 6 patients withdrew informed consent and 3 had unresectable tumors intraoperatively in the Exp-group, while in the Con-group, 6 patients withdrew informed consent and 5 had unresectable tumors intraoperatively. All these patients were excluded and finally, 122 patients in the Exp-group and 120 cases in the Con-group were included for modified intention-to-treat analysis. The demographic and clinical characteristics of the included patients were shown in Table 1. There were no statistical differences between the two groups in age, BMI, ECOG status, ASA grading, comorbidities, ratio of neoadjuvant/postoperative adjuvant chemotherapy, and tumor characteristics.

### Urogenital functions
According to protocol, patients undergoing Non-$R_0$ resection or abdominal perineal resection (APR) were excluded for urogenital function assessment, thus 107 cases in the Exp-group and 100 cases in the Con-group were included for per-protocol analysis. The result was shown in Supplementary Tables 1 and 2. Briefly, compared with the Con-group, the Exp-group revealed better outcomes of RUV (POW2, POM3, and POM6), Max-UFR (POW2 and POM6), IPSS (POW2 and POW3), and urinary dysfunction rate (POW2), respectively. For erectile and ejaculation functions, the Exp-group continuously revealed better IIEF5 score and lower ejaculation dysfunction rate even until 12 months postoperatively. This result was consistent with previously published initial outcomes[16].

### Postoperative recovery and surgical outcomes
As shown in Supplementary Table 3 and 4, there were no statistical differences in postoperative recovery data and surgical outcomes between the two groups, containing first flatus, first liquid or semi-liquid diet, removal of drainage tube and catheter, surgical time, intraoperative bleeding, methods of surgical procedure and rate of a defunctioning stoma. No cases of conversion or 30-day mortality occurred. The total incidence of perioperative complications was 17.8% (n = 43/242), containing 22 cases (18.0%) in the Exp-group and 21 cases (17.5%) in the Con-group. The most common complication was

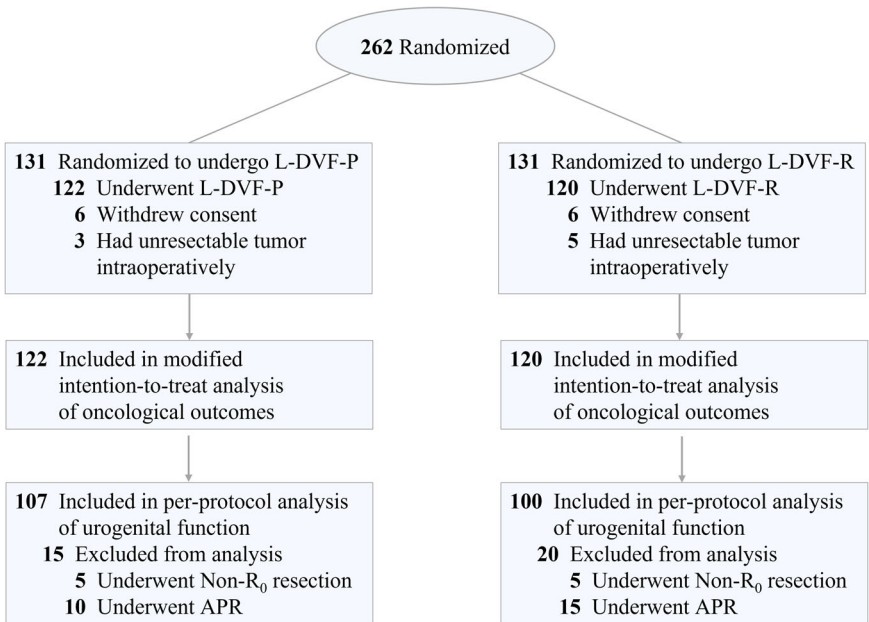

**Fig. 1 | CONSORT diagram, including enrollment and outcomes.** L-DVF-P, laparoscopic total mesorectal excision with Denonvilliers' fascia preservation; L-DVF-R, laparoscopic total mesorectal excision with Denonvilliers' fascia resection.

**Table 1 | Demographic and clinicopathological characteristics of patients in the modified intention-to-treat population**

| Characteristic | Exp-group (n = 122) | Con-group (n = 120) |
|---|---|---|
| Age (years) | 57.8 ± 8.4 | 58.2 ± 8.7 |
| BMI (kg/m²) | 22.3 ± 3.0 | 22.7 ± 3.4 |
| **ECOG performance status** | | |
| 0 | 84 (68.9%) | 85 (70.8%) |
| 1 | 38 (31.1%) | 35 (29.2%) |
| **ASA grading** | | |
| I | 78 (63.9%) | 74 (61.7%) |
| II | 37 (30.3%) | 38 (31.7%) |
| III | 7 (5.7%) | 8 (6.7%) |
| **Comorbidities** | | |
| None | 81 (66.4%) | 74 (61.7%) |
| ≥1 | 41 (33.6%) | 46 (38.3%) |
| **Neoadjuvant chemotherapy** | | |
| Yes | 33 (27.0%) | 35 (29.2%) |
| No | 89 (73.0%) | 85 (70.8%) |
| **Postoperative adjuvant therapy** | | |
| Capecitabine | 10 (8.2%) | 8 (6.7%) |
| CapeOX | 51 (41.8%) | 49 (40.8%) |
| mFOLFOX6 | 11 (9.0%) | 9 (7.5%) |
| None | 50 (41.0%) | 54 (45.0%) |
| Tumor size (cm) | 3.5 ± 1.3 | 3.3 ± 1.5 |
| **Tumor location** | | |
| Anterior | 28 (23.0%) | 33 (27.5%) |
| Lateral | 57 (46.7%) | 50 (41.7%) |
| Posterior | 37 (30.3%) | 37 (30.8%) |
| Tumor height (cm) | 7.4 ± 2.3 | 7.5 ± 2.4 |
| Proximal margin (cm) | 10.1 ± 2.1 | 10.1 ± 1.6 |
| Distal margin (cm) | 3.1 ± 1.2 | 3.0 ± 1.3 |
| Retrieved lymph nodes (No.) | 19.0 ± 9.7 | 17.2 ± 8.4 |
| Metastatic lymph nodes (No.) | 1.3 ± 4.0 | 1.7 + 3.9 |
| **Histology** | | |
| Differentiated | 109 (89.3%) | 105 (87.5%) |
| Poorly differentiated | 13 (10.7%) | 15 (12.5%) |
| **TME grading classification** | | |
| I | 118 (96.7%) | 116 (96.7%) |
| II | 4 (3.3%) | 4 (3.3%) |
| III | 0 | 0 |
| **Pathologic T stage** | | |
| 1 | 14 (11.5%) | 16 (13.3%) |
| 2 | 24 (19.7%) | 27 (22.5%) |
| 3 | 40 (32.8%) | 32 (26.7%) |
| 4 | 44 (36.1%) | 45 (37.5%) |
| **Pathologic N stage** | | |
| 0 | 78 (63.9%) | 68 (56.7%) |
| 1 | 30 (24.6%) | 30 (25.0%) |
| 2 | 14 (11.5%) | 22 (18.3%) |
| **Pathologic TNM stage** | | |
| I | 33 (27.0%) | 35 (29.2%) |
| II | 45 (36.9%) | 33 (27.5%) |
| III | 44 (36.1%) | 52 (43.3%) |

*BMI* body mass index, *ECOG* Eastern Cooperative Oncology Group, *ASA* American Society of Anesthesiologists.

anastomotic leakage (8.0% in the Exp-group and 7.6% in the Con-group, respectively).

### Overall survival and disease-free survival

At the last follow-up, 40 patients (16.5%) had died, the median follow-up time was 51.9 months. Among them, 4 cases died from non-tumor-related diseases, containing 2 cases of cardiovascular deaths, 1 case of stroke, and 1 case of traffic accident. Taking together, there were 36 cases (14.9%) of tumor-related deaths in this study (18 cases in the Exp-group and 18 cases in the Con-group). The overall survival (OS) was calculated and shown in Fig. 2. The 3-year OS was 94.1% in the Exp-group and 89.7% in the Con-group (Log-rank $P = 0.22$; hazard ratio [HR], 0.56; 95% CI, 0.22–1.42). In detail, the 3-year OS for the Exp-group and Con-group was 96.9% vs. 94.3% in Stage I (Log-rank $P = 0.60$; HR, 0.53; 95% CI, 0.05–5.80), 95.4% vs. 90.7% in Stage II (Log-rank $P = 0.46$; HR, 0.51; 95% CI, 0.08–3.04) and 90.7% vs. 85.8% in Stage III (Log-rank $P = 0.45$; HR, 0.62; 95% CI, 0.18–2.13).

We set truncation at 36 months for restricted mean survival time (RMST). The RMST in the Exp-group was 35.50 months (95% CI, 35.05–35.94 months). Correspondingly, the restricted mean times lost (RMTL) was 0.50 months (95% CI, 0.06–0.95 months). In contrast, the RMST was 34.95 months (95% CI, 34.21–35.69 months) and the RMTL was 1.05 months (95% CI, 0.32–1.79 months) in the Con-group. The RMST ratio for L-DVF-P to L-DVF-R was 1.02 (95% CI, 0.99–1.04), suggesting that the Exp-group had a mean survival time of 2% more than that of the Con-group ($P = 0.21$). Univariate analysis of OS at 3 years revealed no differences in outcomes of the two groups in terms of tumor location or stage (Supplementary Table 5).

The disease-free survival (DFS) was presented in Fig. 3. The 3-year DFS was 87.5% in the Exp-group and 85.6% in the Con-group (Log-rank $P = 0.64$; HR, 0.85; 95% CI, 0.42–1.67). In detail, the 3-year DFS for the Exp-group and Con-group was 93.8% vs. 94.3% in Stage I (Log-rank $P = 0.94$; HR, 1.08; 95% CI, 0.15–7.66), 88.5% vs. 87.9% in Stage II (Log-rank $P = 0.86$; HR, 0.89; 95% CI, 0.24–3.31) and 81.8% vs. 78.0% in Stage III (Log-rank $P = 0.68$; HR, 0.82; 95% CI, 0.33–2.05). Univariate analysis of DFS at 3 years revealed no differences in outcomes of the two groups in terms of tumor location or stage (Table 2).

### Recurrence rate and pattern

At the last follow-up, 37 patients (15.3%) were diagnosed with recurrence, containing 9 cases of local recurrence (24.3%), 12 cases of liver metastasis (32.4%), 13 cases of lung metastasis (35.1%) and 3 cases of peritoneal metastasis (8.1%). As shown in Table 3, the recurrence rate in the Exp-group and Con-group was 15.6% ($n = 19$) and 15.0% ($n = 18$) respectively, and the difference was not significant.

### Discussion

Urogenital dysfunction has become the major complication of total mesorectum excision (TME) for low-mid rectal cancer. In this study, we revealed that compared with traditional TME surgery, TME surgery with Denonvilliers' fascia (DVF) preservation had better postoperative urogenital function, with a comparable oncological outcome, thus may be a better choice for male rectal cancer patients with specific staging.

It was reported that more than 50% of patients treated for rectal cancer experienced a deterioration in sexual function, while urinary dysfunction occurred in one-third of patients[17]. Severe urinary dysfunction is rare because it usually could be ameliorated 3–6 months postoperatively[18]. In this study, the incidence of urinary dysfunction in the Con-group was 25.7% two weeks postoperatively, much higher than that of the Exp-group (6.3%). However, at either 3 or 6 months postoperatively, the incidence of urinary dysfunction decreased obviously and reached as low as 5.8% in the Con-group. A multi-modal study examining long-term urogenital function after rectal cancer surgery also revealed that only 7.8% of patients reported their bladder habits to be a moderate or big problem[19]. However, nerve-injury-related sexual dysfunction was considered difficult to be ameliorated. In this study, the incidence of erectile dysfunction in the Con-group still reached as high as 39.0% at 12 months postoperatively. Meanwhile, the incidence of ejaculation dysfunction also did not decrease significantly within 12 months postoperatively. This result was consistent with previous long-term study, which revealed 36.2% of sexual dysfunction after

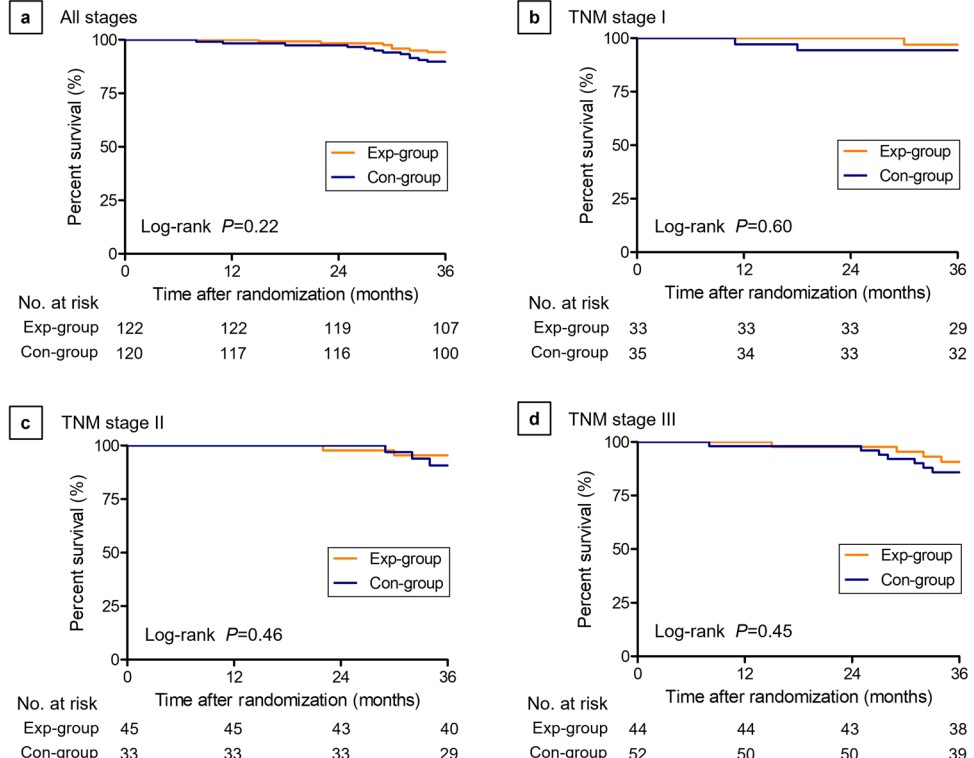

**Fig. 2 | Overall Survival (OS) for laparoscopic total mesorectal excision (TME) with Denonvilliers' fascia preservation (Exp-group) vs laparoscopic TME with Denonvilliers' fascia resection (Con-group) at 3 years after surgery.** Kaplan–Meier method was used to estimate survival probabilities over time and the log-rank test was applied to compare survival curves between two groups. **a** Patients with all stages of cancer. **b** Patients with TNM stage I cancer. **c** Patients with TNM stage II cancer. **d** Patients with TNM stage III cancer.

TME[19]. Although radiotherapy was also considered to have a role in the development of sexual dysfunction, pelvic autonomic nerve (PAN) damage during TME was universally acknowledged as the main cause of urogenital dysfunction. Thus, modifying surgical procedures of TME has gained great attention from colorectal surgeons.

Previous studies have proved that DVF acted as a protective sheet for PAN and thus partial resection of DVF may lead to PAN injury and postoperative urogenital dysfunction[5,10,15]. Nevertheless, traditional TME surgery performed anterior to DVF with partial resection of DVF is still widely performed by most colorectal surgeons, because it used to be considered no surgical plane posterior to DVF and thus entire preservation of DVF was technically difficult and impracticable[8].

Figuring out an appropriate surgical procedure is the key to solve this problem. There used to be two surgical procedures for anterior dissection. First, dissection 1–1.5 cm above peritoneal reflection, which helps better exposure of the anterior pelvic cavity, is especially beneficial for patients with obesity or narrow pelvic cavity. Second, dissection at the lowest level of peritoneal reflection. In clinical practice, it was difficult to dissect posterior to DVF with the first surgical procedure. However, dissection at the lowest level of peritoneal reflection sometimes helps enter the surgical plane posterior to DVF. Based on this, we performed both cadaveric study and surgical video review, finding that DVF began with a white thickened line at the lowest level of peritoneal reflection, and ended at the perineal body[13]. Thus, this white thickened line can be considered a surgical marker of DVF. Due to intraoperative traction and counter traction on the rectum, the fusion of the fascia is mobile and not always located at the lowest level of peritoneal reflection, which can explain why dissection at the lowest level of peritoneal reflection does not always help enter posteriorly to DVF. On the contrary, the surgical marker of DVF is immobile, thus dissection below this marker line leads to entry posterior to DVF easily,

regardless of mobilization of the peritoneal reflection. With the help of this surgical line, dissection posterior to DVF becomes feasible and practicable, and thus DVF could be preserved entirely[12,13].

Some may still doubt the general applicability of DVF preservation, especially for high BMI patients. Although the mean BMI was normal in this study, there were also some cases of overweight and obesity, while the procedure was performed smoothly regardless of the high BMI. Usually, for patients with high BMI or narrow pelvic cavity, hanging the peritoneal reflection with a suture or performing traction of the rectum with tieback will help better exposure of both the pelvic cavity and the anterior wall of rectum, and thus make surgery easier to generalize.

Although previous studies have strongly indicated that DVF resection may be the reason for PAN injury, there is still a lack of clinical trial studies to prove the effect of DVF preservation on urogenital function protection and confirm the oncological safety. Based on this, we conducted the PUF-01 study with two aims. First, to evaluate the advantage of preserving DVF during laparoscopic TME on protecting postoperative urogenital function in male patients with rectal cancer. Both the initial results[16] and updated per-protocol analysis in this study proved that compared with partly resection of DVF, preservation of DVF presented lower incidences of urinary and sexual dysfunctions. Second, to verify the oncological safety of DVF preservation. The results revealed that the 3-year OS (94.1% vs. 89.7%), 3-year DFS (87.5% vs. 85.6%) and recurrence rate (15.6% vs. 15.0%) between DVF preservation and resection groups were all similar. In addition, the 3-year RMST ratio for L-DVF-P to L-DVF-R was 1.02, suggesting that the Exp-group had a mean survival time of 2% more than that of the Con-group. Thus, from a clinical aspect, both survival and DFS data between the two groups were comparable and the result was optimistic.

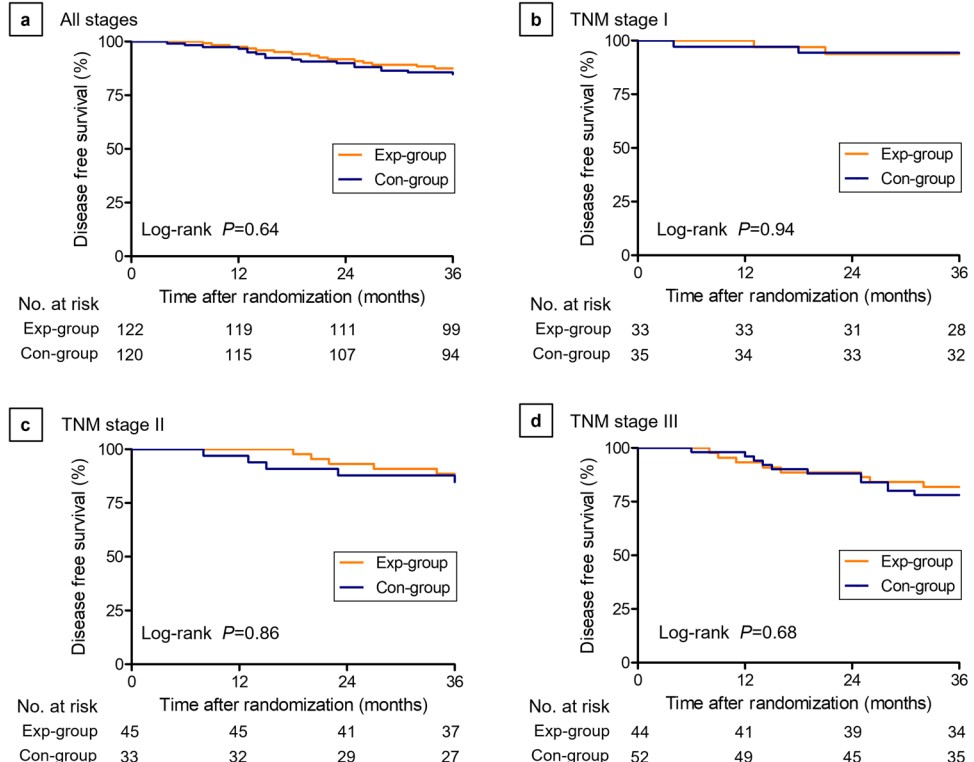

**Fig. 3 | Disease Free Survival (DFS) for laparoscopic TME with Denonvilliers' fascia preservation (Exp-group) vs laparoscopic TME with Denonvilliers' fascia resection (Con-group) at 3 years after surgery.** Kaplan–Meier method was used to estimate survival probabilities over time and the log-rank test was applied to compare survival curves between two groups. **a** Patients with all stages of cancer. **b** Patients with TNM stage I cancer. **c** Patients with TNM stage II cancer. **d** Patients with TNM stage III cancer.

### Table 2 | Univariate analysis of disease-free survival at 3 years' follow-up

| Variable | Patients No. | Exp-group, 3-y DFS (95% CI), % | Patients No. | Con-group, 3-y DFS (95% CI), % | Hazard ratio[a] | Log-rank P value |
|---|---|---|---|---|---|---|
| Total | 122 | 87.5(81.7–93.6) | 120 | 85.6(79.5–92.2) | 0.85(0.42–1.67) | 0.640 |
| **T stage** | | | | | | |
| $T_{1-2}$ | 38 | 94.6(87.6–100.0) | 43 | 90.7(82.4–99.8) | 0.57(0.10–3.09) | 0.511 |
| $T_{3-4}$ | 84 | 84.3(76.8–92.5) | 77 | 82.7(74.5–91.7) | 0.88(0.41–1.91) | 0.754 |
| **N stage** | | | | | | |
| $N_0$ | 78 | 90.8(84.5–97.5) | 68 | 91.2(84.7–98.2) | 1.01(0.34–3.01) | 0.985 |
| $N_1$ | 30 | 86.7(75.3–99.7) | 30 | 72.5(58.0–90.7) | 0.47(0.14–1.56) | 0.218 |
| $N_2$ | 14 | 70.7(50.2–99.6) | 22 | 85.7(72.0–100.0) | 2.23(0.50–9.98) | 0.294 |
| **TNM stage** | | | | | | |
| I | 33 | 93.8(0.04–85.7) | 35 | 94.3(86.9–100.0) | 1.08(0.15–7.66) | 0.939 |
| II | 45 | 88.5(79.5–98.5) | 33 | 87.9(77.4–99.8) | 0.89(0.24–3.31) | 0.861 |
| III | 44 | 81.8(71.1–94.0) | 52 | 78.0(67.4–90.4) | 0.82(0.33–2.05) | 0.677 |
| **Tumor location** | | | | | | |
| Anterior | 28 | 96.3(89.4–100.0) | 33 | 90.9(81.6–100.0) | 0.39(0.04–3.74) | 0.413 |
| Non-anterior | 94 | 84.9(77.9–92.5) | 87 | 83.5(76.0–91.8) | 0.90(0.43–1.89) | 0.778 |

[a]Reference, Con-group.

In this study, cases of APR or Non-R$_0$ resection were excluded for per-protocol analysis of postoperative urogenital function. This is because that APR may have an influence on postoperative urogenital function. Also, Non-R$_0$ resection usually requires additional radiotherapy, which also has potential adverse effects on urogenital function. Nevertheless, these patients were still included in the modified intention-to-treat analysis for oncological outcomes.

Considering that tumor location and T stage may have an impact on DVF preservation and tumor recurrence, we performed a univariate analysis of 3-year DFS and OS based on variables containing tumor location and T stage. The results revealed no differences in either OS or DFS of $T_{1-2}$ or $T_{3-4}$, anterior wall, or other locations between the Exp- and Con-group. However, we should also keep in mind that since this was the first and explorative RCT study on DVF preservation, only $T_{1-2}$ patients were included for rectal cancer located in the anterior wall. Further study may be performed to investigate whether preservation of DVF is also suitable and oncological safe for $T_3$ or even $T_{4a}$ anterior rectal cancer.

Neoadjuvant radiotherapy or chemoradiotherapy was believed to help control local recurrence of locally advanced rectal cancer[20,21].

**Table 3 | Recurrence rate and pattern of patients in the modified intention-to-treat population**

|                      | Exp-group (n = 122) | Con-group (n = 120) | P value |
|----------------------|---------------------|---------------------|---------|
| Total recurrence     | 19 (15.6%)          | 18 (15.0%)          | 0.901   |
| Local recurrence     | 4 (3.3%)            | 5 (4.2%)            |         |
| Liver metastasis     | 7 (5.7%)            | 5 (4.2%)            |         |
| Lung metastasis      | 6 (4.9%)            | 7 (5.8%)            |         |
| Peritoneal metastasis| 2 (1.6%)            | 1 (0.8%)            |         |

Data are analyzed using Pearson's two-sided $\chi^2$-test.

However, the latest FOWARC Trial also demonstrated that compared to fluorouracil with radiotherapy, neoadjuvant mFOLFOX6 chemotherapy without radiotherapy revealed similar oncological outcomes and fewer adverse reactions for patients with locally advanced rectal cancer[22]. In this study, we did not include patients with neoadjuvant radiotherapy because of the potential adverse effects of radiotherapy on urogenital function. The total incidence of local recurrence in this study was 3.7%, with a median follow-up of 51.9 months, and the local recurrence was comparable between the L-DVF-P and L-DVF-R group, suggesting that even without neoadjuvant radiotherapy, laparoscopic TME with DVF preservation was of oncological safety for locally advanced rectal cancer.

This study had several limitations. First, patients with neoadjuvant radiotherapy were not included in this trial. Further studies should be performed to investigate whether laparoscopic TME with DVF preservation is also feasible, as well as of better postoperative urogenital function and comparable oncological outcome for them. Second, some cases in this study did not reach the follow-up of 60 months. Thus, the 5-year OS, DFS, and recurrence rate should be furthered follow-up to get a more convincing result of oncological safety. Third, for rectal cancer located in the anterior wall, only $T_{1-2}$ patients were included in this study. Further study may be performed to investigate whether preservation of DVF is also suitable and oncological safe for $T_3$ or even $T_{4a}$ anterior rectal cancer.

In conclusion, the PUF-01 trial revealed that laparoscopic TME with DVF preservation was feasible and safe, had the advantage on postoperative urogenital function, as well as comparable 3-year OS and DFS oncological results, thus may be a better choice for male rectal cancer patients with specific staging.

## Methods

### Study design
The PUF-01 trial is an open-label, multicenter, randomized clinical trial conducted at 11 centers in China. The trial was registered on Clinical-Trials.gov on April 26, 2015 (https://clinicaltrials.gov/study/NCT02435758) and conducted according to the Helsinki Declaration of 1975. The protocol (Supplementary Note) was approved by the Ethics Committee of the Third Affiliated Hospital, Sun Yat-Sen University.

### Participants
Patients were enrolled from August 26, 2015, through May 6, 2020. Eligibility criteria were listed in (Supplementary Table 6). Briefly, male patients were included if they were aged 20 to 71 years, pathological diagnosis of rectal adenocarcinoma with tumors from anal edge 6–12 cm (the distance was measured routinely by rigid proctoscope and digital rectal examination); preoperative staging $T_{1-4}$ ($T_{1-2}$ for anterior rectal wall) $N_{0-2}M_0$ (AJCC-7th, Pelvic MRI, chest and abdominal CT scan were mandatory for staging); $R_0$ resection is expected; had an Eastern Cooperative Oncology Group (ECOG) performance status of 0 (asymptomatic) or 1 (symptomatic but completely ambulatory); preoperative American Society of Anaesthesiologists (ASA) grade I–III; preoperative normal urinary function (bladder residual urine volume, RUV <100 ml), normal erection function (5-item version of the International Erectile Function Index Questionnaire, IIEF-5 > 21) and ejaculation function grading as I level. Patients were excluded if tumors were with extensive invasion of surrounding tissues or imaging examination in regional integration intumescent lymph nodes (maximum diameter 3 cm or higher). All candidates provided written informed consent.

There were two reasons why only male patients were enrolled in this study. First, the structure of DVF was more complicated and multiple-morphologic for females, thus the feasibility of DVF preservation for females was still unclear. Second, the assessment method of sexual function for females was relatively insufficient.

### Randomization and blinding
This study is an open-label and a single-blind design is adopted in this study. Stratified blocked randomization was used, the stratification factor was center, and the block size was 4. The random allocation sequence was generated by a statistician who was independent of the research, using the SAS 9.3 software (SAS Institute, Cary, NC). Participants were randomly assigned using random envelopes in a 1:1 ratio to groups that underwent laparoscopic TME with DVF preservation (L-DVF-P, Exp-group) or DVF resection (L-DVF-R, Con-group), respectively. The surgeons were informed of grouping information preoperatively, while the participants and research assistants enrolling in patient follow-up and functional evaluation were blinded.

### Interventions
Following randomization, laparoscopic TME surgery was performed. In the Con-group, dissection of the anterior rectum was performed anterior to DVF, and the fascia was resected by an "inverted U-shaped" incision ≥2 cm beneath the tumor. In contrast, for the Exp-group, dissection was performed posterior to DVF and thus DVF was preserved entirely[16].

To ensure the surgical homogeneity and quality, video recordings of each procedure were stored for reference, and mandatory intraoperative photographs of specific fields to verify PAN protection were obtained illustrating: (1) the area of ligation of the inferior mesenteric artery, (2) the area of bilateral hypogastric nerve, (3) bilateral rectal ligament area, (4) the anterior rectal wall and DVF area. The integrity of the gross specimen and histopathological examination for TME grading classification were evaluated[23]. Meanwhile, for the Con-group, to confirm the histopathology of the DVF, a histopathological examination was performed on the DVF-covered 10-2 o'clock area of the mesorectum[5].

Postoperative prophylactic antibiotics and pain medications, fluid therapy, and nutritional support were administered in accordance with routine medical practice. Adjuvant chemotherapy was arranged if needed, using capecitabine, CapeOX, or mFOLFOX6 regimen.

### Outcome measures
The patients' urinary function was evaluated by RUV (mL, by ultrasonography), maximum flow rate (MFR, mL/s, by urodynamics), and International Prostate Symptom Score (IPSS). Erectile function and ejaculation function were evaluated by IIEF-5 and ejaculation function grading (Grade I: normal ejaculation; Grade II: retrograde ejaculation; Grade III: anejaculation), respectively. The initial results of postoperative urogenital function have been published in the previous paper[16].

The oncological data included the 3-year overall survival (OS) and disease-free survival (DFS), and recurrence rate. Postoperative follow-up was performed every 3 months within 2 years, and every 6 months 3–5 years postoperatively. Comprehensive hematology, chest and

abdomen spiral CT, and colonoscopy were used to evaluate the patient's postoperative survival status. OS was calculated from the day of randomization until the day of death (event) or the day of the last follow-up examination (censored), while DFS was calculated from the day of randomization until the day of recurrence or death (event) or the day of the last follow-up examination (censored). Data were censored for patients with no evidence of diseases at the last follow-up examination or for patients who died from other diseases or reasons without evidence of recurrence. The last follow-up was on October 24, 2022.

## Sample size calculation

In this study, the incidences of urinary dysfunction 2 weeks postoperatively and sexual dysfunction 12 months postoperatively were the primary endpoints and dominant evaluation indicators. In our previous study, the incidence of urinary dysfunction and sexual dysfunction were 24.39% and 9.76%, respectively, for DVF-preserving procedures; the corresponding incidences for DVF-resecting procedures were 44.68% and 42.55%[24]. According to the superiority study design, the sample size was determined using an alpha of 5% as the unilateral statistical significance level, setting the power of the test to 90%. The final sample size takes the maximum 1 from the 2 indicators. At least 110 patients were required in each group. The sample size was calculated using the PASS 15.0 software.

## Statistical analysis

DFS and OS were evaluated by the Kaplan–Meier method and compared by the log-rank test. The Cox proportional hazards regression model was used to estimate the adjusted hazard ratios (HRs) and 95% confidence intervals (CIs) for the effect of surgical approach on DFS and OS. The "survminer" package in R was used to provide various functions for survival analysis, including testing for differences in OS between groups. The median follow-up time was calculated using the reverse Kaplan–Meier method. Restricted mean survival time (RMST) was used to quantify the survival time, and the RMST ratio and 95% CI were obtained by survRM2 package in R software[25]. To construct a 95% CI, we estimated the asymptotic variance of RMST and formed the CI by RMST ± 1.96 (estimated standard deviation). For inference of the ratio type metrics, we used the delta method to calculate the standard error. Specifically, we considered $\log\{\mu^\tau(1)\}$ and $\log\{\mu^\tau(0)\}$, and calculated the standard error of log-RMST. We then calculated a CI for log-ratio of RMST, and transformed it back to the original ratio scale. If the 95% CI was relatively tight around 0, it suggested that the difference in RMST had no statistical significance. Either OS or DFS of laparoscopic TME with DVF preservation was considered to be noninferior to DVF resection with the 1-sided 95% confidence interval (CI) and a margin for a hazard ratio (HR) of 1.34. The margin of HR was discussed and determined by PUF Research Collaboration Group and statistician, based on previous studies[26,27]. The data were expressed as mean ± standard deviation (SD) for continuous variables and frequency for categorical variables. Quantitative data were analyzed using the $t$-test; qualitative data, Pearson's or Cochran-Mantel-Haenszel $\chi^2$-test; rank data, nonparametric test. $P$-values < 0.05 were considered as statistically different. Statistical analysis was performed using the SPSS 25.0 statistical software (IBM Corp., USA) and R, version 3.6.2 (R Group for Statistical Computing).

## Reporting summary

Further information on research design is available in the Nature Portfolio Reporting Summary linked to this article.

## Data availability

De-identified and processed participant data will be shared beginning 3 months and ending 5 years following publication by requesting the corresponding author (Hongbo Wei, E-mail: weihb@mail.sysu.edu.cn) for academic purposes. The corresponding author will reply to the request within 2 months, subject to the approval of the ethics committees of the Third Affiliated Hospital, Sun Yat-Sen University. Source data underlying Figs. 2, 3 are provided with this paper. The study protocol is available as a supplementary file (Supplementary Note). Source data are provided with this paper.

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

## Acknowledgements

We thank Professor R.J. Heald for his guidance and generous help during the study. We thank Hao Chen (Nanfang Hospital of Southern Medical University, China) for the consultation. We thank Dr. Yang Shuo and Dr. Luo Hao (Department of Epidemiology and Biostatistics, School of Public Health; Sun Yat-Sen University), Xiaohua Li (the Third Affiliated Hospital of Sun Yat-sen University, China) for the statistical consultation. This study was supported by grants from the Sun Yat-Sen University Clinical Medicine Research 5010 Program (No. 2015016), the National Natural Science Foundation of China (No. 81971378, No. 81901471) and the Natural Science Foundation of Guangdong Province (No. 2021A1515010577, No. 2022A1515012653). The funder of the study had no role in study design, data collection, data analysis, data interpretation, or writing of the report.

## Author contributions

J.F. and H.W. contributed to design and writing. B.W., Z.Z., J.Z., J.L. X.Y., T.C., Y.H., J.H., and J.L. contributed acquisition, analysis, and interpretation of data. J.X., F.H., M.H., Q.X., X.W., C.H., G.W., Y.J., G.S., and H.D. contributed surgical support.

## Competing interests

The authors declare no competing interests.

## Additional information

[1]Department of Gastrointestinal Surgery, The Third Affiliated Hospital, Sun Yat-sen University, 600 Tianhe Road, Guangzhou, People's Republic of China. [2]Department of Gastrointestinal Surgery, Anyang Cancer Hospital, the Fourth Affiliated Hospital, Henan University of Science and Technology, 1 Huanbin North Road, Anyang, People's Republic of China. [3]Department of Gastrointestinal Surgery, Sun Yat-sen Memorial Hospital, Sun Yat-sen University, 107 Yanjiang West Road, Guangzhou, People's Republic of China. [4]Department of Colorectal Surgery, The Sixth Affiliated Hospital, Sun Yat-sen University, 26 Yuancun Erheng Road, Guangzhou, People's Republic of China. [5]Department of Gastrointestinal Surgery, Affiliated Hospital of Guangdong Medical University, People's Avenue, Zhanjiang, People's Republic of China. [6]Department of Gastrointestinal Surgery, Shantou Central Hospital, Waima Road, Shantou, People's Republic of China. [7]Department of Gastrointestinal Surgery, the Second Affiliated Hospital, Guangzhou Medical University, 250 Changgang East Road, Guangzhou, People's Republic of China. [8]Department of Gastrointestinal Surgery, the First Affiliated Hospital, Henan University of Science and Technology, 636 Guanlin Road, Luoyang, People's Republic of China. [9]Department of Gastrointestinal Surgery, Shunde Hospital of Southern Medical University, 1 Licun Jiazi Road, Foshan, People's Republic of China. [10]Department of Gastrointestinal Surgery, the First Affiliated Hospital, Xiamen University, 55 Zhenhai Road, Xiamen, People's Republic of China. [11]Department of General Surgery, Nanfang Hospital of Southern Medical University, 1838 Guangzhou Avenue North, Guangzhou, People's Republic of China. [12]Department of Medical Statistics, School of Public Health, Sun Yat-sen University, 74 Zhongshan Second Road, Guangzhou, People's Republic of China. [13]Chinese PUF Research Collaboration Group Center, 600 Tianhe Road, Guangzhou, People's Republic of China. [14]These authors contributed equally: Jiafeng Fang, Bo Wei, Zongheng Zheng. ✉e-mail: weihb@mail.sysu.edu.cn

## Chinese Postoperative Urogenital Function (PUF) Research Collaboration Group

Jiafeng Fang [1,14], Bo Wei[1,14], Zongheng Zheng[1,14], Jian'an Xiao[2], Fanghai Han[3], Meijin Huang[4], Qingwen Xu[5], Xiaozhong Wang[6], Chuyuan Hong[7], Gongping Wang[8], Yongle Ju[9], Guoqiang Su[10], Haijun Deng [11] & Hongbo Wei [1] ✉

