## [Peer Review File · Nature Communications]

Reviewers' Comments:

Reviewer #1:

Remarks to the Author:

This is a well conducted randomized controlled trial comparing preservation or en bloc resection of Denonvilliers' fascia in male patients with rectal cancer that is not located anteriorly or confined to the rectal wall (max. T2) if located on the anterior side. None of the patients had preoperative radiotherapy. The results show better urogenital function if preserving Denonvilliers' fascia, which is also what I would have expected beforehand. The manuscript is of good textual quality.

The authors state that traditional TME requires partly resection of Denonvilliers' fascia, but in my view this is debatable and not generally acknowledged anymore. The rationale of performing this trial in the current era is not completely clear to me, because TME surgery is often individualized based on tumour location and extent since we have detailed information from preoperative MRI, aiming at an optimal balance between oncological safety and nerve preservation. Therefore, most surgeons will not resect Denonvilliers' fascia if the tumour is located on the dorsal or dorsolateral side, and this might even be safe if there is at least 2 mm margin on MRI to this fascia in anteriorly located tumours. The authors confirm that preserving Denonvilliers' fascia, if oncologically safe, gives the best outcomes. But in my opinion, this is common practice, at least in my country, and the trial does not provide any practice changing results. This discussion is especially relevant, since only a quarter of included patients had an anteriorly located tumour (Table 1). Therefore, the issue of preservation of Denonvilliers' fascia was only clinically relevant in a minor subgroup.

There is an important issue related to extrapolation of these results to other countries and patient populations. Western populations might be different due to obesity and different anatomy, and more application of neoadjuvant radiotherapy.

The two types of surgery are described as straight forward, but in reality it might sometimes be quite difficult to clearly identify Denonvilliers' fascia and to choose the plane either ventral or dorsal to this fascia. This is also due to high variability in appearance of Denonvilliers' fascia in different patients, as pointed out by the authors in the discussion. The authors describe video registration and photographs for quality control of the surgical interventions, but do not provide any results for assessment of video's or pictures.

Some detailed comments:

Title: it is not clear from the title that this is an RCT

Discussion:

This is too long and might benefit from more focus on the main findings of the study and comparing their results with urogenital function in other studies that included these outcomes, not necessarily looking at DVF preservation.

The detailed anatomical discussion is probably something for a narrative review, but not as part of an original manuscript on an RCT.

The authors state that it would be interesting to also look at anterior T3 and T4 tumours, but this seems impossible or even unethical, as DVF is at least threatened or even involved in those patients. Please explain.

The fact that female patients were not included is not a limitation, because the DVF is not a clearly defined structure in women, and therefore the research question is not valid for this population.

Reviewer #2:

Remarks to the Author:

Inclusion of Denonvilliers' fascia became a standard in total mesorectal excision (TME) for rectal cancer. The primary aim was to improve the oncological outcome in the area with the open surgical approach. With the laparoscopic approach the visibility of the anatomic structures has improved with the possibilities of more accurate dissection that may preserve the Denonvilliers'

fascia and reducing the risk of traumatizing the neurogenetic innervation of the bladder and sexual function in men.

In this elegant and very well-designed randomized study the results showed that the preservation of Denonvilliers' fascia significantly improved the sexual and urinary functional outcome without compromising the oncological outcome. This indeed is new and important knowledge with a high impact on the surgical treatment of rectal cancer in men. Inclusion and exclusion criteria are well-described and relevant. A strength is that those health care professionals that investigated the primary outcomes were blinded to the intervention performed. Thus, the results are very reliable. The statistical methods are appropriate. Another a strength is that professor Heald (the "father of TME") has been a member of the international advisor group in this study. The manuscript is very well-written.

The only thing a mis is information how many patients that were included at each participating center and whether all patients planned for a low anterior resection for curative intention were evaluated and included in the "group of eligible patients".

Reviewer #3:

Remarks to the Author:

This paper presents the analysis of secondary outcomes (OS and DFS) of a randomized clinical trial evaluating DVF preservation versus resection. The primary analysis regarding the effect on postoperative urogenital function was previously published. There are several flaws in the analysis and inconsistencies in the presented results that are of major concern and limit the interpretation of the results as presented. These are listed in detailed below.

Major concerns:

1. The results in the text do not match the survival graphs and eTable5 presented. This is of major concern. The survival graphs sample size also don't match Table 1. For example, the log-rank p-values are different and the sample sizes are different for the stratified analysis.
2. The analysis is stated as intention to treat, but it is per protocol. In an intention to treat analysis all randomized subjects should be analyzed in this case all 262. The per-protocol analysis should be a secondary analysis.
3. Both the OS and DFS analyses should include all deaths as events since these are not disease specific mortality, but overall survival.
4. DFS should include both recurrence and death as events. Since probability of disease-free survival is being estimated, death can't be censored.
5. For the primary analysis, DFS should start from randomization.
6. RMST graphs are not needed for OS since the curves should be the same as the KM graphs, but do not seem to be the same.
7. The statistical analysis section does not include many of the analysis performed and the references for them. For example:
 - a. The methods for the estimation of the HR are not specified.
 - b. The methods for obtaining the 95%CI in line 154
 - c. The method for the p-value in line 155.
 - d. Line 155, univariate analysis of OS at 3 year, what test was used?

Minor concerns:

1. Specify how median follow-up time was estimated.
2. In line 154, should it be mean survival time instead of median?
3. In line 353, what is the unilateral statistical significance versus the alpha?
4. In line 354, do you mean power instead of efficiency?
5. X-axis for OS KM should be time from randomization.

Dear Editor and Reviewers,

Thank you very much for your comments concerning our manuscript entitled “Effect of Intraoperative Denonvilliers’ Fascia Preservation on Urogenital Function Protection and Oncological safety for Rectal Cancer” (Manuscript ID: NCOMMS-23-15046A). Those comments are all valuable and very helpful for revising and improving our paper, as well as the important guiding significance to our research. We have studied the comments carefully and the followings are our responses to the reviewer point by point. We also upload a revised version where the revisions were marked in yellow.

Responses to the reviewer’s comments:

Reviewer #1:

1. The authors state that traditional TME requires partly resection of Denonvilliers' fascia, but in my view this is debatable and not generally acknowledged anymore. The rationale of performing this trial in the current era is not completely clear to me, because TME surgery is often individualized based on tumour location and extent since we have detailed information from preoperative MRI, aiming at an optimal balance between oncological safety and nerve preservation. Therefore, most surgeons will not resect Denonvilliers' fascia if the tumour is located on the dorsal or dorsolateral side, and this might even be safe if there is at least 2 mm margin on MRI to this fascia in anteriorly located tumours. The authors confirm that preserving Denonvilliers' fascia, if oncologically safe, gives the best outcomes. But in my opinion, this is common practice, at least in my country, and the trial does not provide any practice changing results. This discussion is especially relevant, since only a quarter of included patients had an anteriorly located tumour (Table 1). Therefore, the issue of preservation of Denonvilliers' fascia was only clinically relevant in a minor subgroup.

Response: Thanks for the extremely professional comment. We strongly agree that TME should be performed individualized based on tumor location and extent. However, this opinion has not been widely accepted, especially in China and other Asian countries. In 2004, R.J. Heald described a holy plane for TME surgery and demonstrated that there was usually no surgical plane posterior to Denonvilliers’ fascia (DVF). Thus, the

optimal TME for rectal cancer was by dissection anterior to DVF and a U-shaped cut of DVF should be performed to avoid damage of bilateral neurovascular bundles (NVB) [1]. Since Professor Heald is honored as “father of TME”, seldom surgeons doubt with his opinion, and thus traditional TME with partly DVF resection was still routinely performed in many large-scale medical centers of China even when the tumor was located on the dorsal or dorsolateral side, or T1-2 anterior wall. When we presented the potential necessity and advantage of TME surgery with DVF preservation in large academic conferences previously, many colorectal surgeons doubted it. In fact, there are still researches and multicentre RCT in progress to investigate the necessity of TME surgery with routine partly resection of DVF [2-4]. Thus, we believe the surgical approach of TME is still largely controversial.

In addition, although some surgeons, like you and us, agree that DVF should not be routinely resected in TME surgery, there is still no RCT so far to confirm the effect of DVF preservation on urogenital function preservation compared to partly resection in TME surgery. Thus, we believe that this RCT study is in urgent need to provide solid evidence on advantage of TME with DVF preservation and thus clarify the appropriate surgical ideas of TME. Although only a quarter of included patients had an anteriorly located tumor, this study also enrolled patients with T₁₋₄N₀₋₂M₀ un-anterior tumor, and proved that TME with DVF preservation was not only oncological safe, but also of urogenital function advantage for all enrolled patients. Thus, we believed that the issue of DVF preservation was not only clinically relevant in a minor subgroup. We have added some of these statements which were marked in yellow in the Introduction Section of revised paper. Specially thanks for your professional comment.

References for this statement

- [1] Heald RJ, Moran BJ, Brown G, Daniels IR. Optimal total mesorectal excision for rectal cancer is by dissection in front of Denonvilliers' fascia. *Brit J Surg* 2004; 91: 121-3.
- [2] Zheng Z, Ye D, Wang X, Lu X, Huang Y, Chi P. Effect of partial preservation versus complete preservation of Denonvilliers' fascia on postoperative urogenital function in male patients with low rectal cancer (PREDICTION): protocol of a multicentre,

prospective, randomised controlled clinical trial. *BMJ Open*. 2022;12(4):e055355.

[3] Pan C, Xiaojie W. Membrane anatomy: motivation to promote precise laparoscopic and robot colorectal surgery. *Chin J Gastrointest Surg*. 2019;22(5):406-412.

[4] Ghareeb WM, Wang X, Chi P, Wang W. The “multilayer” theory of Denonvilliers’ fascia: anatomical dissection of cadavers with the aim to improve neurovascular bundle preservation during rectal mobilization. *Colorectal Dis Off J Assoc Coloproctology G B Irel*. 2020;22(2):195-202.

2. There is an important issue related to extrapolation of these results to other countries and patient populations. Western populations might be different due to obesity and different anatomy, and more application of neoadjuvant radiotherapy.

Response: Thanks for the professional comment. It is true that patients from different countries may differ from characteristics and anatomy. In this study, we also found that different patients presented different morphologies of either DVF or surgical landmark line. As presented below, the surgical videos revealed different morphology of DVF in Figure 1R. Some were thinner while some may be thicker. Some were gray while others may be light red. Also, Figure 2R presented the surgical landmark line located at the lowest level of peritoneal reflection. Some cases of this white line were thinner while some were thicker. Nevertheless, this surgical line existed persistently, and as we illustrated in the manuscript, regardless of either mobilization of the peritoneal reflection or different morphology of DVF, dissection below this marker line resulted in easily entry posterior to DVF. Thus, we believe that different anatomy will not restrict the extrapolation of the results in this trial.

Figure 1R. Different morphology of DVF

Figure 2R. Different morphology of surgical landmark line (shown as white dot line)

Although the mean BMI was normal in this study, there were also some cases of overweight (n=64) and obesity (n=6), while the procedure was performed smoothly regardless of the high BMI. Usually, for patients with high BMI, hanging the peritoneal reflection with suture (Figure 3R) or performing traction of the rectum with tieback (Figure 4R) will help better exposure of either pelvic cavity or the anterior wall of rectum, and thus make surgery easier to generalize. Also, as we discussed previously, the surgical marker line also existed persistently in patients with high BMI. Thus, DVF preservation with the guidance of this surgical marker line is also applicable in patients with obesity. We previously discussed with some colorectal surgeons from Japan, Korea and western countries, and they have also performed laparoscopic TME with DVF preservation for benign colorectal diseases or early-stage rectal cancers. Thus, we believe that this surgical procedure is generalizable in either China or other countries.

Figure 3R. Hanging the peritoneal reflection with suture to help exposure of the pelvic cavity

Figure 4R. Traction by tieback to help exposure of the pelvic cavity

We didn't enroll cases of neoadjuvant radiotherapy in this study because of the potential adverse effects of radiotherapy on urogenital function. In our clinical experience, cases with neoadjuvant radiotherapy, if needed, still could be performed TME with DVF preservation smoothly. Usually, for cases of neoadjuvant radiotherapy, electric hook was better than ultrasonic scalpel when performing pelvic dissection. Nevertheless, as we discussed in the manuscript, we totally agree that further trial should be conducted to assess the feasibility and advantage of TME with DVF preservation for patients with neoadjuvant radiotherapy.

3. The two types of surgery are described as straight forward, but in reality it might sometimes be quite difficult to clearly identify Denonvilliers' fascia and to choose the plane either ventral or dorsal to this fascia. This is also due to high variability in appearance of Denonvilliers' fascia in different patients, as pointed out by the authors in the discussion. The authors describe video registration and photographs for quality control of the surgical interventions, but do not provide any results for assessment of videos or pictures.

Response: Thanks for the professional comment. It's such a good question that not only the professional reviewer, but also some experts doubt with this issue. In our very early stage of laparoscopic TME surgery, we usually dissected 1-1.5cm above peritoneal reflection when performing anterior dissection, and found that it was extremely difficult

to dissect posterior to DVF. On the contrary, dissection at the lowest level of peritoneal reflection sometimes help enter the surgical plane posterior to DVF. We were confused why dissection on this procedure did not always lead to an appropriate surgical plane. Then, through both cadaveric study and surgical videos review, we found that DVF began with a white thickened line at the lowest level of peritoneal reflection, and ended to the perineal body (Figure 5R).

Figure 5R. DVF originated from the lowest level of peritoneal reflection and formed as a thickened line. SV: seminal vesicle

In other words, this white thickened line can be considered as surgical marker of DVF. Due to intraoperative traction and countertraction on the rectum, the fusion of the fascia was mobile and not always located at the lowest level of peritoneal reflection, which can explain why dissection at the lowest level of peritoneal reflection not always help enter posteriorly to DVF. On the contrary, the surgical marker of DVF is immobile, thus dissection below this marker line leads to entry posterior to DVF easily, while dissection above this marker line leads to entry anterior to DVF, regardless of mobilization of the peritoneal reflection (Figure 6R). With the help of this surgical line, identification of DVF and selection of two different procedures of anterior dissection

has become feasible and practicable.

All intraoperative photographs of specific fields were reviewed and assessed to verify PAN protection, containing the anterior plane. All cases presented good PAN preservation. In the Exp-group, the DVF was well preserved, while DVF was partly resected as protocol in the Con-group. We have added some of these statements in the revised manuscript.

Figure 6R. Two procedures for anterior wall dissection basing on the surgical marker line. A1-A3, dissection below marker line led to entry posterior to DVF and then DVF could be entirely preserved. B1-B3, dissection 1-1.5cm above marker line would directly enter the surgical plane anterior to DVF and DVF would be partly resected.

4. Title: it is not clear from the title that this is an RCT.

Response: Thanks for the comment. According to the reviewer's suggestion, as well as due to the *Nature Communications* formatting instructions (no more than 15 words for the Title), we would like to revised the title to "Effect of Denonvilliers' Fascia Preservation on Male Urogenital Function and Oncological Safety (the PUF-01 Study)".

5. Discussion: This is too long and might benefit from more focus on the main findings of the study and comparing their results with urogenital function in other studies that included these outcomes, not necessarily looking at DVF preservation. The detailed anatomical discussion is probably something for a narrative review, but not as part of an original manuscript on an RCT.

Response: Thanks for the suggestion. We discussed anatomy in the present manuscript in order to clarify some misunderstandings of DVF and thus help readers better understand the meaning of this paper. Nevertheless, according to the reviewer's suggestion, we have simplified contents of Discussion Section. Also, we compared and discussed the incidence of urogenital dysfunction in this manuscript with other previous studies, which was marked in yellow in the first paragraph of Discussion Section. We wish that the revised Discussion Section would be more precise and of better quality.

6. The authors state that it would be interesting to also look at anterior T3 and T4 tumors, but this seems impossible or even unethical, as DVF is at least threatened or even involved in those patients. Please explain.

Response: Thanks for the professional comment. It's true that for anterior T3 and T4 tumors, it may not be suitable for dissection posterior to DVF due to threaten of tumor invasion. However, because DVF doesn't belong to mesorectum, it is not located within the "holy plane" and thus should not be resected even for T3 or T4a anterior tumor in theory. Whether preservation of DVF is suitable and possible for T3 or even T4a is controversial and unclear, so it may be further investigated in the future. Nevertheless, we absolutely agree that this clinical trial should be carefully assessed with ethical agreement before being conducted.

7. The fact that female patients were not included is not a limitation, because the DVF is not a clearly defined structure in women, and therefore the research question is not valid for this population.

Response: Thanks for the suggestion. We totally agree that the structure of DVF is more complicated and multiple-morphologic for female. In addition, the assessment method of sexual function for female is relatively insufficient. Thus, we did not enroll female rectal cancer patients in this study. According to the reviewer's suggestion, we have deleted the limitation description of female patients in the manuscript. In addition, we added the word "Male" in the Title of manuscript to follow the rules in the 'Sex and Gender Equity in Research-SAGER-guidelines' of the journal.

Special thanks for your professional comments and suggestions.

Reviewer #2:

1. The only thing a mis is information how many patients that were included at each participating center and whether all patients planned for a low anterior resection for curative intention were evaluated and included in the “group of eligible patients”.

Response: Thanks very much for the reviewer’s affirmation to our work. Please find attached the patient distribution among each participating center. Meanwhile, in most of the participating medical centers, patients planned for a low anterior resection for curative intention were routinely evaluated, then patients who met the inclusion criteria were included in the “group of eligible patients”.

No.	Participating center	Cases
1	The Third Affiliated Hospital, Sun Yat-sen University	132
2	The Sixth Affiliated Hospital, Sun Yat-sen University	19
3	Nanfang Hospital of Southern Medical University	4
4	Anyang cancer Hospital, Henan University of Science and Technology	31
5	Affiliated Hospital of GuangDong Medical University	18
6	Shantou Central Hospital	13
7	The Second Affiliated Hospital, Guangzhou Medical University	8
8	Shunde Hospital of Southern Medical University	5
9	The First Affiliated Hospital, Xiamen University	8
10	Sun Yat-sen Memorial Hospital, Sun Yat-sen University	18
11	The First Affiliated Hospital, Henan University of Science and Technology	6
	Total	262

Reviewer #3:

1. The results in the text do not match the survival graphs and eTable5 presented. This is of major concern. The survival graphs sample size also don't match Table 1. For example, the log-rank p-values are different and the sample sizes are different for the stratified analysis.

Response: We apologize for this mistake and confusion. First, we really appreciate the profession and rigor of the reviewer. These comments precisely hit the nail, thus greatly help us improve our manuscript and avoid stupid writing mistakes. In our previous manuscript, we did not include some cases of patients with APR surgery, thus some data were mixed by mistake and thus confusing. To avoid this, this time we invited two individual statisticians (Dr. Yang Shuo and Dr. Luo Hao from Department of Epidemiology and Biostatistics, School of Public Health, Sun Yat-Sen University) to analyze all data again and double check the revised version to be correct, containing Figure 2, Figure 3, Table 1, Table 2 and eTable 5. We are sorry for the statistic mistakes. Also, in the Acknowledgments Section, we thank Dr. Yang Shuo and Dr. Luo Hao for their contribution of data re-analysis.

For the survival graphs (Figure 2 and 3), we drew the survival curve within 5 years in order to present more survival information, thus the log-rank p-values represented the statistic results of 5-year OS or DFS in Figure 2/3. While in the text, we presented the results of 3-year OS and DFS. That's why the log-rank p-values were different between the Figure and text description. Please find attached the 3-year OS and DFS graphs below, and if the reviewer consider it is better to use a same standard to describe the oncologic outcome, we would be willing to replace the Figure 2 and 3 to demonstrate the survival curve within 3 years.

Figure R1 Overall Survival for Exp-group vs Con-group at 3 years after surgery.

Figure R2 Disease Free Survival for Exp-group vs Con-group at 3 years after surgery.

2. The analysis is stated as intention to treat, but it is per protocol. In an intention to treat analysis all randomized subjects should be analyzed in this case all 262. The per-protocol analysis should be a secondary analysis.

Response: Thanks for the professional comment. It's true that for an ITT analysis, all randomized subjects should be analyzed. In this study, as shown in Figure 1, totally 262 patients were enrolled and randomly assigned to the Exp-group or Con-group (n=131 respectively). However, there were 9 cases in the Exp-group and 11 cases in the Con-group who accepted randomization but didn't finish laparoscopic rectal cancer resection (withdrew consent or had unresectable tumor intraoperatively). Thus, refer to previous studies [1,2], we used modified intention-to-treat analysis (mITT) for the remaining 122 cases in the Exp-group and 120 cases in the Con-group. We believe that mITT should be more objective than ITT to assess the survival outcome between the two groups in this case. To make it more precise, we have revised the description in the manuscript from ITT to mITT.

According to protocol, patients undergoing non-R₀ resection or abdominal perineal resection (APR) were excluded for urogenital function assessment, thus 107 cases in the Exp-group and 100 cases in the Con-group were included for per-protocol analysis. Thanks for the reviewer's kindly and professional suggestion.

References for this statement

- [1] Alexander E, Goldberg L, Das AF, et al. Oral Lefamulin vs Moxifloxacin for Early Clinical Response Among Adults with Community-Acquired Bacterial Pneumonia: The LEAP 2 Randomized Clinical Trial. *JAMA*. 2019 Nov 5;322(17):1661-1671.
- [2] Huang C, Liu H, Hu Y, et al. Laparoscopic vs Open Distal Gastrectomy for Locally Advanced Gastric Cancer: Five-Year Outcomes From the CLASS-01 Randomized Clinical Trial. *JAMA Surg*. 2022 Jan 1;157(1):9-17.

3. Both the OS and DFS analyses should include all deaths as events since these are not disease specific mortality, but overall survival.

Response: Thanks for the professional suggestion. We do agree that both the OS and DFS analyses should include all deaths as events. Thus, we have re-analyzed the data

and redrawn the Figure 2 and 3. The revised results were marked in yellow in the revised manuscript and we wish that it would be more precise.

4. DFS should include both recurrence and death as events. Since probability of disease-free survival is being estimated, death can't be censored.

Response: Consistent with the above reply, we agree that DFS should include both recurrence and death as events. We have changed death from censored to events and re-analyzed the data. Thanks for your kindly comment.

5. For the primary analysis, DFS should start from randomization.

Response: Thanks for reminding. We have corrected the description of DFS both in the text (Paragraph of Outcome measures in the Methods Section) and in the Figure 3, and also updated the data of DFS.

6. RMST graphs are not needed for OS since the curves should be the same as the KM graphs, but do not seem to be the same.

Response: Thanks for the comment. We have deleted the RMST graph from Figure 2 according to the suggestion, while the data of RMST were still preserved in the text. Meanwhile, we double checked the RMST graphs and considered it to be the same as the KM graphs. The reason why it seemed that they were not the same may be ascribed to be different ordinate starting scale of Y-axis between the two graphs (The KM starts from 0% while the RMST starts from 50%).

7. The statistical analysis section does not include many of the analysis performed and the references for them. For example:

a. The methods for the estimation of the HR are not specified.

Response: We used Cox proportional hazard models to estimate hazard ratios (HRs) for quantifying the influence of various functional variables. The noninferiority margin for a hazard ratio (HR) of 1.34 was discussed and determined by Chinese Postoperative Urogenital Function (PUF) Research Collaboration Group and statistician, based on

previous studies [1,2]. We have added this statement in the Statistical analysis Section of revised manuscript, marked in yellow.

References for this statement

[1]Fujita S, Mizusawa J, Kanemitsu Y, et al. Mesorectal Excision With or Without Lateral Lymph Node Dissection for Clinical Stage II/III Lower Rectal Cancer (JCOG0212): A Multicenter, Randomized Controlled, Noninferiority Trial. Ann Surg. 2017 Aug; 266(2):201-207.

[2] Royston P, Parmar MK. Restricted mean survival time: an alternative to the hazard ratio for the design and analysis of randomized trials with a time-to-event outcome. BMC Med Res Methodol. 2013 Dec 7;13:152.

b. The methods for obtaining the 95% CI in line 154

Response: The RMST ratio and 95% CI was obtained by survRM2 package in R software [1]. To construct a 95% confidence interval, we estimated the asymptotic variance of RMST and formed the confidence interval by $RMST \pm 1.96$ (estimated standard deviation). We have added brief statement in the revised manuscript.

References for this statement

[1]Zhou M. Restricted mean survival time and confidence intervals by empirical likelihood ratio. J Biopharm Stat. 2021 May 4;31(3):362-374.

c. The method for the p-value in line 155.

Response: For inference of the ratio type metrics, we used the delta method to calculate the standard error. Specifically, we considered $\log\{\mu^{\tau(1)}\}$ and $\log\{\mu^{\tau(0)}\}$, and calculated the standard error of log-RMST. We then calculated a confidence interval for log-ratio of RMST, and transformed it back to the original ratio scale. If the 95% CI was relatively tight around 0, it suggests that the difference in RMST has no statistical significance.

d. Line 155, univariate analysis of OS at 3 year, what test was used?

Response: We use the "survminer" package in R to provide various functions for

survival analysis, including testing for differences in overall survival (OS) between groups. A statistical significance test for OS was conducted using the "survminer" package, following the survdiff() function or the coxph() function. The survdiff() function performs the log-rank test, which is a common nonparametric test for comparing survival distributions between groups.

8. Specify how median follow-up time was estimated.

Response: Thanks for the question. The median follow-up time was the median observation time to the event or last follow-up. It was calculated using the reverse Kaplan-Meier method in this study. We have added the statement in the revised manuscript.

9. In line 154, should it be mean survival time instead of median?

Response: Sorry for the writing mistake. It should be mean survival time; we have made correction in the revised manuscript.

10. In line 353, what is the unilateral statistical significance versus the alpha?

Response: Sorry for the writing mistake. We have corrected this statement to "According to the superiority study design, the sample size was determined using an alpha of 5% as the unilateral statistical significance level, setting power of the test to 90%".

11. In line 354, do you mean power instead of efficiency?

Response: Thanks for the kindly remind. We have corrected "efficiency" to "power".

12. X-axis for OS KM should be time from randomization.

Response: Thanks for the professional suggestion. We have redrawn the OS KM graph and corrected the X-axis to "Time after randomization (months)".

Special thanks for your professional comments and suggestions.

We tried our best to improve the manuscript and made changes in the revised manuscript. These changes will not influence the content and framework of the paper. And here we did not list the changes but marked in yellow in revised paper. We appreciate for Editors/Reviewers' warm work earnestly, and hope that the correction will meet with approval.

Once again, thank you very much for your comments and suggestions.

Yours sincerely,

Hongbo Wei

E-mail: weihb@mail.sysu.edu.cn

2023.07.09

Reviewers' Comments:

Reviewer #1:

Remarks to the Author:

I would like to thank the authors for thoroughly answering the comments and making the appropriate corrections to their manuscript.

Considering the revised version, I have some minor comments:

1. At the end of the introduction, the authors now have already summarized the main findings of their study. In my view, the last paragraph of the introduction should only include the aim of the study.
2. The first paragraph of the discussion should shortly summarize the main findings of the study. Subsequently the findings should be compared with relevant literature in the second paragraph. In the revised manuscript, the discussion now starts with some general statements about urogenital function. So, I would propose to slightly rewrite the first part of the discussion.
3. The quality of the English writing should be improved with the help of a native English speaker.

Reviewer #3:

Remarks to the Author:

The revised paper and analysis have addressed most of the previous comments. The only remaining items that should be further clarified are:

1. I am still unclear why the log rank is different between the 3 and 5 year OS and DFS since the data is the same and the log rank test is used to compare the entire curve not a single time point. The analysis (Fig 2 and 3, and Table 2) should be based on all available follow-up as of the data lock for the comparison of OS and DFS. This way the text and Figures will match. The estimates of the probability of OS and DFS at specific time points can be reported along with their 95% CI even if the data does not end there. If patients are censored at 3 years that should be clarified in the methods.

In the interpretation, the authors should clarify that the logrank test and HR estimate are for the comparison of the OS and DFS curve, and not a specific test for the 3 year OS and DFS which the text seems to indicate.

2. All the statistical methods and references for the RMST and the log rank that were included in the response to reviewer's comments should also be included in the Methods section of the paper for clarity and reproducibility. Only some were included.

Dear Editor and Reviewers,

Thank you very much for your second-round comments concerning our manuscript entitled “Effect of Denonvilliers’ Fascia Preservation on Male Urogenital Function and Oncological Safety (the PUF-01 Study)” (Manuscript ID: NCOMMS-23-15046A). As previously, we have studied the comments carefully and the followings are our responses to the reviewer point by point. We also upload a revised version where the revisions were marked in yellow.

Responses to the reviewer’s comments:

Reviewer #1:

1. At the end of the introduction, the authors now have already summarized the main findings of their study. In my view, the last paragraph of the introduction should only include the aim of the study.

Response: Thanks for the suggestion. We summarized the main findings of this study at the end of the Introduction Section to comply with the *Nature Communications* formatting instructions, which suggested that “The results of the current study must only be discussed in the final paragraph of Introduction”. Nevertheless, if the reviewer and editor feel it’s better to delete this statement and just include the aim of the study in this paragraph, we would be willing to revise it.

2. The first paragraph of the discussion should shortly summarize the main findings of the study. Subsequently the findings should be compared with relevant literature in the second paragraph. In the revised manuscript, the discussion now starts with some general statements about urogenital function. So, I would propose to slightly rewrite the first part of the discussion.

Response: Thanks for the suggestion. We have rewritten the first part of the discussion, shortly summarized the main finding of this study, and then discussed the incidence of urogenital dysfunction both in this study and in the literatures in the subsequent paragraphs.

3. The quality of the English writing should be improved with the help of a native English speaker.

Response: According to the reviewer's suggestion, we have invited a native English speaker to help improve the English writing of this paper. We will upload a revised version of the manuscript with improved English writing. Also, if the journal thinks it's necessary to improve the English writing through the help of some language editing company, we will be glad to do that.

Special thanks for your professional comments and suggestions.

Reviewer #3:

1. I am still unclear why the log rank is different between the 3 and 5 year OS and DFS since the data is the same and the log rank test is used to compare the entire curve not a single time point. The analysis (Fig 2 and 3, and Table 2) should be based on all available follow-up as of the data lock for the comparison of OS and DFS. This way the text and Figures will match. The estimates of the probability of OS and DFS at specific time points can be reported along with their 95% CI even if the data does not end there. If patients are censored at 3 years that should be clarified in the methods.

Response: We are sorry for the confusion about the log-rank test for 3- and 5-year survival outcomes. It's true that the log-rank should be individual and indicate the survival curve, but not a specific time point. In the text, the result of log-rank was for the 3-year survival curve, but not for the 3-year time point of the 5-year survival curve. However, in the figures, we presented a 5-year survival curve. That's why the long rank result was different between the text and figure (log-rank test for 3-year curve in the text, but log-rank test for 5-year curve in the figure).

The reason why we presented the 3-year survival result in the text was that the secondary outcomes for this study were 3-year OS and 3-year DFS. Nevertheless, for rectal cancer patients, it is usually necessary to keep follow-up for 5 years. Thus, we kept follow-up of patients enrolled in this study even when they reached the secondary endpoint and drew the 5-year OS and DFS survival curve.

Unfortunately, this made confusion that log ranks varied for different timepoint of the survival curve, and the texts and figures did not match. To avoid confusion and misunderstanding, we drew the 3-year OS and 3-year DFS survival curve, replaced the 5-year survival curve, and presented them in the revised Figure 2 and 3, as presented below. Then, all data (containing log-rank test and survival curves in Figure 2 and 3, the description in the text, Table 2, etc.) will match and thus avoid confusion.

Thanks for your patience and professional suggestion, which strongly improve our manuscript and make it much clearer and more precise.

Figure 2: OS for TME with DVF preservation (Exp-group) vs DVF resection (Con-group) at 3 years after surgery.

Figure 3: DFS for TME with DVF preservation (Exp-group) vs DVF resection (Con-group) at 3 years after surgery.

2. In the interpretation, the authors should clarify that the log rank test and HR estimate are for the comparison of the OS and DFS curve, and not a specific test for the 3 year OS and DFS which the text seems to indicate.

Response: According to the reviewer's suggestion, in the Method Section, we added the following statement: DFS and OS were evaluated by the Kaplan-Meier method and compared by the log-rank test. The Cox proportional hazards regression model was used to estimate the adjusted hazard ratios (HRs) and 95% confidence intervals (CIs) for the effect of surgical approach on DFS and OS.

3. All the statistical methods and references for the RMST and the log rank that were included in the response to reviewer's comments should also be included in the Methods section of the paper for clarity and reproducibility. Only some were included.

Response: Thanks for the suggestion. We have added the contents of statistical methods and references for the RMST and the log-rank which were included in the response to reviewer's comments to the Methods section of the revised paper, which as marked in yellow.

Special thanks for your professional comments and suggestions.

We really appreciate for Editors/Reviewers' warm work earnestly, and hope that the correction will meet with approval. Once again, thank you very much for your comments and suggestions.

Yours sincerely,

Hongbo Wei

E-mail: weihb@mail.sysu.edu.cn

2023.08.17

Reviewers' Comments:

Reviewer #3:

Remarks to the Author:

The authors have addressed the comments.